# Cyclosporines Antagonize the Antiviral Activity of IFITMProteins by Redistributing Them toward the Golgi Apparatus

**DOI:** 10.3390/biom13060937

**Published:** 2023-06-03

**Authors:** David Prikryl, Mariana Marin, Tanay M. Desai, Yuhong Du, Haian Fu, Gregory B. Melikyan

**Affiliations:** 1Department of Pediatrics, Division of Infectious Diseases, Emory University School of Medicine, Atlanta, GA 30322, USA; 2Children’s Healthcare of Atlanta, Atlanta, GA 30322, USA; 3Carl Zeiss Microscopy, White Plains, NY 10601, USA; 4Department of Pharmacology and Chemical Biology, Emory University School of Medicine, Atlanta, GA 30322, USA; 5Emory Chemical Biology Discovery Center, Emory University School of Medicine, Atlanta, GA 30322, USA; 6Department of Hematology and Medical Oncology, Winship Cancer Institute, Atlanta, GA 30322, USA

**Keywords:** cyclosporin, virus fusion, host restriction factors, IFITM trafficking, high-throughput compound screening, beta-lactamase virus fusion assay, confocal microscopy, Golgi

## Abstract

Interferon-induced transmembrane proteins (IFITMs) block the fusion of diverse enveloped viruses, likely through increasing the cell membrane’s rigidity. Previous studies have reported that the antiviral activity of the IFITM family member, IFITM3, is antagonized by cell pretreatment with rapamycin derivatives and cyclosporines A and H (CsA and CsH) that promote the degradation of IFITM3. Here, we show that CsA and CsH potently enhance virus fusion with IFITM1- and IFITM3-expressing cells by inducing their rapid relocalization from the plasma membrane and endosomes, respectively, towards the Golgi. This relocalization is not associated with a significant degradation of IFITMs. Although prolonged exposure to CsA induces IFITM3 degradation in cells expressing low endogenous levels of this protein, its levels remain largely unchanged in interferon-treated cells or cells ectopically expressing IFITM3. Importantly, the CsA-mediated redistribution of IFITMs to the Golgi occurs on a much shorter time scale than degradation and thus likely represents the primary mechanism of enhancement of virus entry. We further show that rapamycin also induces IFITM relocalization toward the Golgi, albeit less efficiently than cyclosporines. Our findings highlight the importance of regulation of IFITM trafficking for its antiviral activity and reveal a novel mechanism of the cyclosporine-mediated modulation of cell susceptibility to enveloped virus infection.

## 1. Introduction

Interferon-induced transmembrane proteins (IFITMs) restrict the entry of a broad range of enveloped viruses, such as the Influenza A Virus (IAV), Vesicular Stomatitis Virus (VSV), Respiratory Syncytial Virus (RSV), and Dengue Virus (DENV). However, Murine Leukemia Virus and arenaviruses, including the Lassa Virus (LASV), are resistant to IFITM restriction [1,2,3,4]. Human genome encodes five IFITM proteins, three of which—IFITM1, IFITM2, and IFITM3—exhibit antiviral activity [5,6,7]. The importance of IFITM-mediated virus restriction in vivo is highlighted by studies showing that *Ifitm3* knockout mice succumb to IAV or RSV infection [8,9,10,11]. Moreover, several studies have shown that single-nucleotide polymorphisms (SNPs) in the *Ifitm3* gene correlate with a higher risk and more severe outcomes of IAV or SARS-CoV-2 infection [12,13,14,15,16]. The IFITMs’ biological functions are not limited to antiviral activities, as they play a role in tumorigenesis (reviewed in [17,18,19]) and can disrupt endosome trafficking and placenta formation [20].

The spectrum of restricted viruses appears to be largely dictated by the IFITM’s subcellular localization, which is modulated by post-translational modifications, including S-palmitoylation [21], ubiquitylation [22,23], phosphorylation [24,25], and methylation [26]. IFITMs are type-II transmembrane proteins [27,28,29,30], although their topology remains controversial [23,31,32,33]. The N-terminal domain of IFITM2 and IFITM3 containing a YEML endocytic sorting motif is a major determinant of their endosomal localization and potent restriction of a number of viruses entering cells via endocytosis [34,35,36,37]. In contrast, IFITM1, which lacks the N-terminal endocytic signal, primarily resides in the plasma membrane (PM) and is more efficient against viruses fusing at this site [2,5,38,39]. Deletion of the first 21 N-terminal residues encompassing the YEML internalization motif of IFITM3 relocalizes this protein to the PM [28,40,41] and shifts the spectrum of restricted viruses to those that are thought to enter at the PM (e.g., Measles virus or SARS-CoV-2) [42,43] that are also targeted by the PM-resident IFITM1 [40]. 

The mechanism of a broad-spectrum restriction of enveloped virus entry by IFITMs is not fully understood. It is believed that different IFITMs inhibit virus fusion through a common mechanism that does not generally involve specific interactions with viral proteins or their cellular receptors, although a few instances of such interactions have been reported [44,45,46]. Current models for IFITM-mediated restriction include (1) rendering the cell membranes more rigid and thereby trapping viral fusion at a hemifusion stage [47,48,49,50,51,52] and (2) accelerating the degradation of incoming viruses by transporting them to lysosomes [53]. In addition to blocking virus entry into cells, IFITMs have been shown to interfere with the spread of newly produced virus particles by incorporating into virions [54,55,56,57]. Finally, IFITMs can inhibit HIV-1 protein synthesis, probably by viral RNA exclusion from polysomes [58]. 

Interestingly, IFITM3′s antiviral activity can be modulated by chemical compounds. It has been reported that analogs of rapamycin (rapalogs) antagonize the antiviral activity of IFITM3 by inducing its degradation through a microautophagy-lysosomal pathway [59,60]. The levels of IFITM3 are also reduced in cells treated with cyclosporine A (CsA) or cyclosporine H (CsH); however, the exact mechanism of IFITM3 degradation by these drugs remains unclear [61].

Here, we carried out a high-throughput screening of compounds that rescue virus fusion with IFITM3-expressing cells and identified cyclosporine A (CsA) as a potent antagonist of IFITM3’s antiviral activity. We show that CsA works by retargeting these proteins from the PM and late endosomes toward the Golgi apparatus but not by promoting IFITM degradation, as has been previously proposed [59,60,61]. The retargeting of IFITMs to the Golgi by CsA and CsH underlies the rescue of the fusion of diverse viruses entering different cell types. Taken together, our results reveal a novel mechanism of regulation of the IFITMs’ activity through their relocation and highlight the importance of understanding the regulation of IFITM trafficking for controlling their adverse effects on essential cellular functions and antiviral activity.

## 2. Materials and Methods

### 2.1. Cell Lines, Plasmids, and Reagents

Human A549, HEK293/17, HeLa, HeLa-IFITM1/2/3 knockout (herein denoted as HeLa-TKO, cells originally from Howard Hang, Rockefeller U, New York, NY, USA), and dog kidney epithelial MDCK cells were obtained from ATCC (Manassas, VA, USA). The cells were grown in the manufacturer’s recommended medium supplemented with 10% heat-inactivated fetal bovine serum (FBS, Atlanta Biologicals, Flowery Branch, GA, USA), 100 U penicillin-streptomycin (Gemini Bio-Products, Sacramento, CA, USA), and the recommended selection of antibiotics when applicable. Stable cell lines A549.IFITM1, A549.IFITM3, and MDCK.IFITM3 cells ectopically expressing respective IFITM proteins have been described previously [47]. HeLa and HeLa-TKO cells ectopically expressing IFITMs were obtained by transducing these with VSV-G-pseudotyped viruses encoding wild-type IFITMs or with the empty vector pQCXIP (Clontech) and selecting with 1.5 µg/mL puromycin.

The pR9ΔEnv, BlaM-Vpr, pcRev, pMDG VSV G, pCAGGS WSN HA and NA, and phCMV-GPc Lassa expression vectors were described previously [47,51,62]. The pMDG-VSV-G expression vector was provided by Dr. John Young (Salk Institute, San Diego, CA, USA). The pCAGGS vectors encoding influenza H1N1 WSN HA and NA were provided by Drs. Donna Tscherne and Peter Palese (Icahn School of Medicine, Mount Sinai, New York, NY, USA) [63]. The LASV GPc plasmid was a gift from Dr. F.-L. Cossett (Université de Lyon, Lyon, France) [64].

Cyclosporine A, Cyclosporine H, recombinant human interferon alpha, Amphotericin B, and cycloheximide were from Sigma (St. Louis, MO, USA). Antibodies used were rabbit anti-IFITM3 against the N-terminus of this protein from Abgent (San Diego, CA, USA), rabbit anti-IFITM3 (Abcam), mouse anti-IFITM2/3 (Proteintech), rabbit anti-IFITM1 (Sigma), mouse anti-GM130 conjugated with AlexaFluor555 from (BD Bioscience, Franklin Lakes, NJ, USA), sheep anti-TGN46 (Bio-Rad, Hercules, CA, USA), and Goat anti-rabbit IgG (H+L) conjugated with AlexaFluor647 (ThermoFisher, Waltham, MA, USA).

### 2.2. Pseudovirus Production and Characterization

To produce pseudoviruses, HEK297T/17 cells were transfected using JetPRIME transfection reagent (Polyplus-transfection, New York, NY, USA). For IAV, LASV, and VSV pseudoviruses (IAVpp, LASVpp, and VSVpp, respectively), ~70% confluent cells in a 100 mm tissue culture dish were transfected with 5 µg pR9deltaEnv, 1.5 µg pMM310, 1 µg pcRev, and with respective envelope glycoprotein-encoding plasmids: pCAGGS WSN HA and NA (2.5 µg each) for IAVpp, 4 µg phCMV-GPc Lassa for LASVpp, and 0.2 µg pMDG VSV G for VSVpp. After 12 h, the transfection medium was replaced with a phenol-red-free growth medium, and cells were cultured for 36 h, at which point the medium was collected, filtered through 0.45 µm PES membrane filter (VWR, Radnor, PA, USA), concentrated 10× using Lenti-X™ Concentrator (Clontech, Mountain View, CA, USA), and stored at −80 °C. The p24/Gag content of the viral stocks was measured using p24 ELISA, as previously described [65].

### 2.3. Virus–Cell Fusion Assay

The β-lactamase (BlaM) assay for virus–cell fusion was carried out, in a modified version of a previously described method [66]. Briefly, 0.5 to 1 ng p24/well of pseudovirus bearing β-lactamase fused to Vpr (BlaM-Vpr) was bound to target cells plated in black clear-bottom 96-well plates by centrifugation at 4 °C for 30 min at 1550× *g*. Unbound viruses were removed by washing, and fusion was initiated by transferring the cells, shifting to 37 °C, 5% CO_2_ for 120 min, after which time cells were loaded with the CCF4-AM BlaM substrate (Life Technologies). The cytoplasmic BlaM activity (ratio of blue to green fluorescence) was measured after overnight incubation at 12 °C, using a Synergy HT fluorescence microplate reader (Agilent Bio-Tek, Santa Clara, CA, USA). Cell viability was determined using the CellTiter-Blue Reagent (Promega); after adding this reagent to cells, the samples were incubated for 30 to 60 min at 37 °C, 5% CO_2_, and cell viability was measured on Synergy HT plate reader (579_Ex_/584_Em_).

### 2.4. LOPAC Library Screen

To identify small-molecule inhibitors of IFITM3 antiviral activity, we used a small library of pharmacologically active compounds (LOPAC) (Sigma-Aldrich). The screening was performed using a BlaM assay in 384-well plates in a robotic platform, as described previously [67]. Briefly, 2 × 10^4^/well cells were dispensed into a 384-well cell culture plate, using Multidrop™ Combi (ThermoFisher), and cultured for 24 h. IAVpp pseudoviruses (0.2 ng p24/well) were dispensed into wells, and the samples were centrifuged at 4 °C for 30 min at 1550× *g*. Next, compounds dissolved in 100% dimethylsulfoxide (DMSO) were dispensed into wells using BeckmanNX liquid handler (Beckman Coulter, Brea, CA, USA) to the final concentration of 20 µM. Each plate contained vehicle control wells that received an equal volume of DMSO (0.4% *v/v*). Samples were incubated at 37 °C, 5% CO_2_ for 120 min, the medium was removed using a BioTek plate washer, and the CCF4-AM substrate was dispensed into the wells. The fusion signal was measured using the EnVision Multilabel plate reader (PerkinElmer, Waltham, MA, USA).

### 2.5. Western Blot Analysis

Cells were harvested and processed, as described elsewhere [66]. Protein bands were detected with rabbit anti-IFITM3, rabbit anti-IFITM1, and mouse anti-GAPDH antibodies (Abcam, Cambridge, MA, USA), and horseradish peroxidase-conjugated Protein G (VWR, Radnor, PA), using a chemiluminescence reagent from GE Healthcare. For cells transduced with IFITM3-expressing vector, only 10% of the amount of cell lysate was used for blotting. The chemiluminescence signal was detected using an XR+ gel doc (Bio-Rad). Densitometry analysis was performed using FIJI [68]. Unless stated otherwise, triplicates were used. Each band was analyzed using the Gel analysis plugin of the FIJI software. DMSO-treated samples were used as reference in each experiment, and the ratio between IFITM3 and GAPDH signals was calculated.

### 2.6. Immunostaining, Microscopy, and Image Analysis

One day prior to imaging, cells were plated in 8-well chamber coverslips (Nunc, Rochester, NY, USA) coated with either 0.1 mg/mL poly-D-lysine (Sigma-Aldrich) or 0.2 mg/mL collagen (Sigma-Aldrich). Cells were treated with indicated compounds/inhibitors or left untreated, fixed with 4% PFA (ThermoFisher) for 20 min at room temperature, permeabilized with 200 µg/mL digitonin for 20 min, and blocked with 10% FBS for 30 min. Cells were next incubated with respective primary antibodies diluted in 10% FBS for 1.5 h, washed, and incubated with secondary antibodies in 10% FBS for 45 min. Samples were stained with Hoechst-33258 (Invitrogen, Waltham, MA, USA) in PBS for 5–10 min prior to imaging. Images were acquired on a Zeiss LSM 880 confocal microscope using a plan-apochromat 63×/1.4 NA oil objective. The entire cell volume was imaged by collecting multiple Z-stacks spaced by 0.41 µm. Images were analyzed using FIJI [68]. Protein signal colocalization (using both Pearson’s and Mander’s coefficients) was computed by the JaCoP FIJI plugin [69] on maximum-intensity projection images.

### 2.7. Statistical Analysis

Unpaired Student’s *t*-test and one-way ANOVA statistical analyses were performed using GraphPad Prism version 9.3.1 for Windows (GraphPad Software, La Jolla, CA, USA). 

## 3. Results

### 3.1. Cyclosporine a Antagonizes the Antiviral Activity of IFITM3

To elucidate the host factors involved in the IFITM3-mediated restriction of IAV, we asked if well-characterized pharmacological agents could promote or antagonize the antiviral activity of IFITM3 and thus reveal host factors that regulate IFITM’s function. Toward this goal, we screened a library of 1280 pharmacologically active compounds (LOPAC, Sigma) for the ability to rescue the fusion of HIV-1 pseudotyped with Influenza A HA (IAVpp) with MDCK cells ectopically expressing IFITM3 (MDCK.IFITM3). IAVpp fusion was measured by a beta-lactamase-based (BlaM) assay that we have previously adapted for high-throughput screening [67]. MDCK cells were chosen because they do not express detectable levels of IFITMs [70]. Control cells transduced with an empty vector (MDCK.Vector) supported robust IAVpp fusion suitable for high-throughput screening. In contrast, IAVpp fusion with MDCK.IFITM3 cells was strongly impaired (Figure 1A and Appendix A). The LOPAC library screening was performed with the final compound concentration of 20 µM. Out of 1280 compounds, we identified a single hit, cyclosporine A (CsA), that potently rescued the IAVpp fusion (Figure 1A). CsA is a natural 11 amino-acid cyclic immunosuppressant polypeptide that binds cyclophilin A (CypA), and the CsA-CypA complex binds to and inhibits the enzymatic activity of calcineurin [71,72].

We next asked whether CsA must be present at the time of virus entry to enhance fusion with IFITM3-expressing cells and whether this effect is cell-type-dependent. Control A549.Vector cells expressing very low levels of endogenous IFITMs [29,47] and A549 cells stably expressing IFITM3 (A549.IFITM3) were pretreated with different concentrations of CsA for 1.5 h at 37 °C, the drug was removed, and cells were infected with IAVpp. CsA pretreatment significantly enhanced IAVpp fusion in A549.IFITM3 cells in a dose-dependent manner, while not having a notable effect on virus fusion with control cells (Figure 1B). 

Petrillo and co-authors [61] have recently reported that CsA and its derivate cyclosporine H (CsH), which does not bind CypA and is not immunosuppressive, enhance VSV-G-pseudotyped lentivirus transduction by antagonizing IFITM3. We therefore tested the effect of CsH on IAVpp fusion and observed a marked increase in fusion with A549.IFITM3 cells but not in A549.Vector cells (Figure 1D). The CsA and CsH concentrations used in fusion experiments did not impact cell viability (Figure 1C,E). These results demonstrate that both CsA and CsH antagonize the antiviral activity of IFITM3. Since CsH does not bind CypA or inhibit calcineurin, its IFITM antagonism should be independent of these interactions.

We next tested if CsA and CsH treatment rescued the fusion of other viruses. For this, we chose Vesicular Stomatitis Virus (VSV) that is thought to fuse with early endosomes and is sensitive to IFITM restriction [41,61], and the Lassa virus (LASV) that fuses with late endosomes and is known to be resistant to IFITM restriction [1,51]. HIV-1 particles pseudotyped with VSV-G (VSVpp) or LASV GPc (LASVpp) glycoproteins were used to infect A549.IFITM3 cells. As expected, pretreatment with CsA or CsH caused a ~10-fold increase in the fusion efficiency of the IFITM3 restriction-sensitive VSVpp, but not LASVpp (Figure 2A,B). In contrast, CsA/CsH treatment had no effect or caused a modest increase in fusion across three different pseudoviruses in the context of control A549.Vector cells, suggesting that cyclosporines do not exert a noticeable non-specific effect on virus fusion and that the fusion rescue is through IFITM antagonism. The similar effects of cyclosporines on viral fusion in MDCK and A549 cells imply that these drugs rescue fusion with IFITM-expressing cells independent of the cell type.

### 3.2. Cyclosporines can Promote Virus Infection without Inducing IFITM3 Degradation

Previous studies of the effects of CsH and different analogs of rapamycin on IFITM3 restriction [59,60,61] employed a prolonged (4–16 h) pretreatment of cells to rescue viral fusion, a condition that was associated with IFITM3 degradation. This is in sharp contrast with our results demonstrating a rescue of viral fusion with IFITM3-expressing A549 cells after a relatively short (1.5 h) pretreatment with CsA or CsH (Figure 2A,B). To address this discrepancy, we treated A549.IFITM3 with CsA, CsH, or DMSO (vehicle) for a short (1.5 h) or long (16 h) time. An analysis of IFITM3 expression by Western blotting revealed no degradation of the ectopically expressed IFITM3 in these cells, regardless of the length of treatment (Figure 2C,D). 

Given that the subcellular localization of IFITMs is critical for their antiviral activity [24,28,40,73], we asked if CsA alters the distribution of IFITM3, which primarily resides in late endosomes [47,53,73,74,75]. The immunostaining of A549.IFITM3 cells for IFITM3 after CsA or CsH treatment showed that, strikingly, these drugs caused a nearly complete relocalization of IFITM3 from endosomes to a perinuclear area (Figure 2E–H). IFITM3 redistribution toward the Golgi was consistently observed using three different anti-IFITM3 antibodies (Appendix A). In control experiments, we pretreated IFITM-expressing A549 cells with the antifungal drug amphotericin B (AmphoB), which is thought to antagonize the antiviral effect of IFITMs by sequestering membrane cholesterol [76]. In contrast to CsA and CsH, AmphoB rescued IAVpp fusion with A549.IFITM1 and A549.IFITM3 cells without causing a notable relocalization of these proteins, as seen by immunostaining (Appendix A). We did not analyze the expression and distribution of ectopically expressed IFITM2 due its homology to IFITM3 resulting in a cross-reactivity of available antibodies. 

**Figure 2 biomolecules-13-00937-f002:**
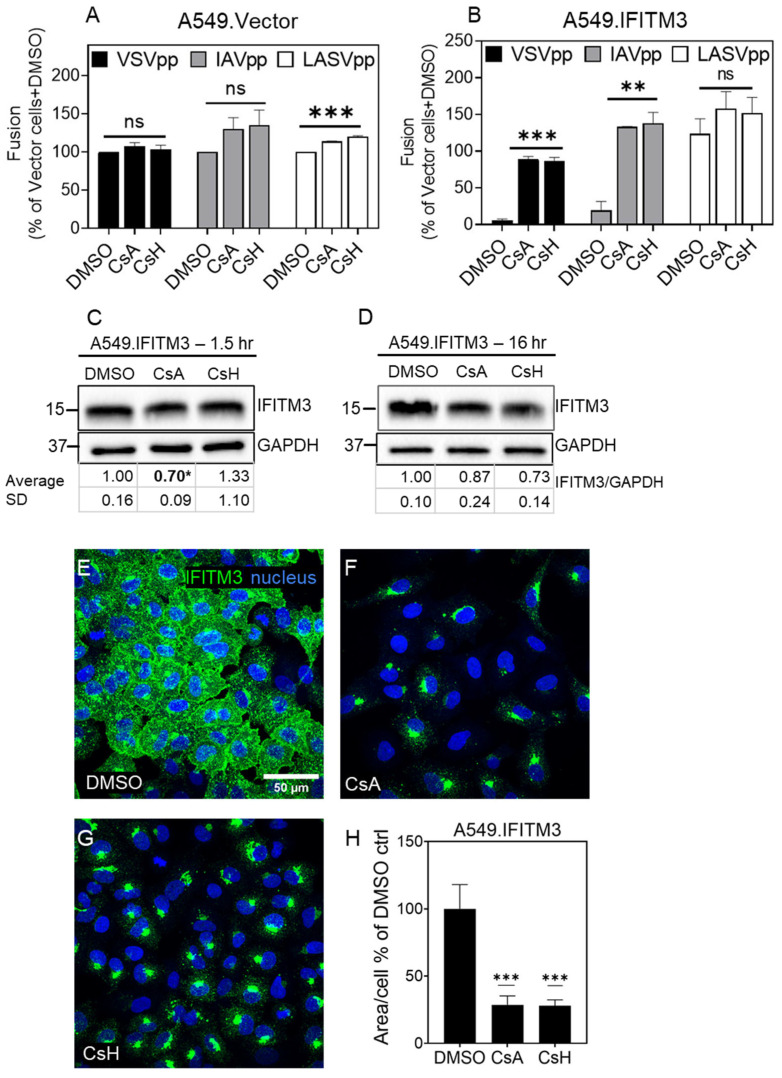
Cyclosporine A rescues virus fusion by relocalizing IFITM3 and not by inducing its degradation. A549.Vector (**A**) and A549.IFITM3 (**B**) cells were incubated with DMSO or 20 µM CsA or CsH for 1.5 h before adding the BlaM-Vpr containing pseudoparticles for virus fusion measurements. Data are means and S.D. of two independent combined experiments, each performed in triplicate. Statistical analysis was performed using one-way ANOVA. For Western blot analysis, A549.IFITM3 were incubated with DMSO or 20 µM CsA or CsH for 1.5 h (**C**) or 16 h (**D**), and total cell lysate was analyzed. Note that Western blots shown in (**C,D**) are representative experiments; the average numbers and S.D. for normalized IFITM band intensities from 3 independent experiments are provided in the tables beneath the blots. The effect of 1.5 h pretreatment with CsA appears significant due to the inclusion of one outlier experiment. A549.IFITM3 cells were incubated with DMSO or 20 µM CsA or CsH for 1.5 h (**E**–**G**), fixed, permeabilized, and stained for IFITM3. For each sample, 3 image fields were acquired. The area covered by IFITM3′s signal per cell was determined from the images using ImageJ, after thresholding, creating an IFITM3-based mask and normalizing to the cell numbers (**H**). Statistical analysis was performed using Student’s *t*-test. * *p* < 0.05; ** *p* < 0.01; *** *p* < 0.001; ns, not significant See also Appendix A.

To further explore the notion that the CsA antagonism of IFITMs is not through the degradation of these proteins, A549.Vector, A549.IFITM1, and A549.IFITM3 cells were treated either with cycloheximide (CHX) to block protein synthesis or a combination of CHX and CsA for 1.5 h. Unlike IFITM3, IFITM1 only modestly inhibited IAVpp fusion (Appendix A), as expected [1]. CsA treatment effectively rescued IAVpp fusion with IFITM3-expressing cells. Importantly, CHX treatment did not significantly affect IAVpp fusion with control or IFITM-expressing cells in the presence or absence of CsA. An analysis of lysates of cells treated with CsA and/or CHX showed no detectable degradation of IFITM3, regardless of the treatment regimen (Appendix A). 

Taken together, our data show that (1) neither IFITM1 nor IFITM3 is quickly degraded in the stable A549 cell lines in the presence of CHX or both CHX and CsA; and (2) a relatively short exposure to CsA that effectively rescues viral fusion does not induce noticeable IFITM3 degradation, even after blocking de novo protein synthesis. These results are in line with the reported IFITM3 turnover time under normal conditions of about 4 h [77]. Thus, IFITM3 degradation does not significantly contribute to the rescue of virus–cell fusion with IFITM3-expressing cells.

### 3.3. Cyclosporine Treatment Quickly Relocates IFITMs toward the Golgi Area

Since IFITM3 removal from the sites of IAV entry (endosomes) is most likely responsible for the rescue of virus fusion in CsA-treated cells, we sought to investigate cellular compartments to which this restriction factor was retargeted. The co-staining of A549.IFITM3 cells with anti-IFITM3 and anti-Golgi antibodies (TGN46 for trans-Golgi, and GM130 for cis-Golgi) [78,79] revealed a markedly increased colocalization of IFITM3 with both Golgi markers in CsA-treated cells compared to the DMSO control (Figure 3). Thus, CsA appears to retarget IFITM3 to the Golgi apparatus. 

We next examined the effect of CsA on IFITM1, which lacks the N-terminal endocytic ^20^YEML^23^ motif of IFITM3 and is therefore primarily localized to the PM [28,43]. CsA treatment relocalized IFITM1 toward the Golgi compartment, similar to the effect of this drug on IFITM3 localization (Figure 4). To further assess the ability of CsA to relocalize IFITMs from the PM, we tested two IFITM3 mutants, the IFITM3 Δ21 (IFITM3 lacking the first 21 N-terminal residues) and the IFITM3 Y20E mutant that lacks the tyrosine-based YEML internalization signal [28,40,41]. As in the case of IFITM1 and wild-type IFITM3, CsA treatment relocalized both mutants to the Golgi (Figure 4). 

Our data suggest that CsA induced a relocalization of IFITM proteins from the PM and endosomes toward the Golgi that does not involve the endocytic sorting motif with the N-terminal domain of IFITM3. Such relocalization appears to restore the ability of viruses to effectively enter IFITM-expressing cells.

We assessed the kinetics of IFITM relocalization toward the Golgi by treating A549.IFITM1 and A549.IFITM3 cells with CsA for varied times and measuring IFITM colocalization with the cis-Golgi marker, GM130, by immunofluorescence. The initial IFITM3 relocalization toward the Golgi was detected in a fraction of cells as early as 15 min after CsA application (Figure 5A). By 30 min, most of the cells exhibited IFITM3 accumulation around the Golgi, with maximum colocalization being achieved by 45 min of treatment (Figure 5B,C,F). A similar, albeit somewhat slower, time course of relocalization toward the Golgi was observed in IFITM1-expressing cells (Figure 5G–L). The difference between the IFITM3 and IFITM1 redistribution dynamics could be due to the slower uptake of IFITM1 from the plasma membrane compared to IFITM3 transport from endosomes toward the Golgi.

### 3.4. Prolonged CsA Exposure induces IFITM3 Degradation in Cells Expressing Low/Endogenous Levels of This Protein

Since IFITM3 is overexpressed in A549.IFITM3 cells, we asked whether CsA and CsH have similar effects on IFITM3 distribution and degradation in HeLa cells that endogenously express IFITMs [70]. Mock-transduced HeLa cells (HeLa.Vector) and cells ectopically expressing IFITM3 (HeLa.IFITM3) were treated with CsA or CsH for 1.5 h and infected with VSVpp, IAVpp, or LASVpp. CsA and CsH markedly enhanced the VSVpp and IAVpp fusion in HeLa.IFITM3 cells, but not in control HeLa.Vector cells (Figure 6A,B). The observed modest (not statistically significant) enhancement of virus fusion with HeLa.Vector cells is most likely due to the endogenous expression of IFITM3 in these cells [76].

Importantly, while a 1.5 h pretreatment with CsA or CsH effectively rescued viral fusion with HeLa cells and caused a dramatic IFITM3 redistribution toward the Golgi, this treatment did not induce a significant IFITM3 degradation (Figure 6C,F). In stark contrast to A549.IFITM3 cells, which did not exhibit notable IFITM3 degradation even after prolonged treatment with cyclosporines (Figure 2C,D), prolonged treatment (4 or 16 h) of HeLa cells strongly reduced the IFITM3 level (Figure 6D,E). 

Since the endogenous expression of IFITM3 in HeLa cells is relatively low [76], we tested if cyclosporines induce IFITM degradation only when it is expressed at low, endogenous levels. Toward this goal, HeLa cells were pretreated for 24 h with interferon-alpha, followed by treatment with CsA or CsH for 1.5 h (Figure 6G) or 4 h (Figure 6H). We observed no IFITM3 degradation in interferon-treated cells upon CsA or CsH treatment for 4 h, suggesting that the cyclosporine-driven IFITM3 degradation in HeLa cells occurs only for the basal, low expression levels of this protein. To verify that interferon treatment itself has no effect on IFITM3 distribution, we immunostained the interferon-alpha-stimulated HeLa cells for IFITM3 and GM130 after pretreatment with CsA or DMSO (Figure 6I). While interferon treatment increased the IFITM3 expression, as expected, interferon did not affect the IFITM3′s subcellular distribution under basal conditions or its relocalization toward the Golgi in the presence of CsA (Figure 6I and Appendix A). 

We sought to further test the notion that the IFITM expression levels in the context of HeLa cells determine the extent of its degradation by cyclosporines. For this purpose, HeLa IFITM1/2/3 triple knockout cells lacking IFITM1, IFITM2, and IFITM3 (hereafter referred to as HeLa-TKO cells) [53] were utilized. As expected, these cells supported efficient VSVpp and IAVpp fusion, which was not affected by cyclosporine treatment (Figure 7A). In comparison, a potent rescue of IAVpp and VSVpp fusion, but not LASVpp fusion, was detected in HeLa-TKO cells ectopically expressing IFITM3 (referred to as Hela-TKO.IFITM3, Figure 7A,B). The treatment of HeLa-TKO.IFITM3 cells with CsA or CsH for a short (1.5 h, Figure 7C) or intermediate (4 h, Figure 7D) length of time did not affect the IFITM3 expression level, as was the case for A549.IFITM3 or interferon-treated HeLa cells (Figure 2C,D and Figure 6F–H). Finally, the immunostaining of HeLa-TKO and HeLa-TKO.IFITM3 for IFITM3 and GM130 revealed a markedly increased colocalization of these proteins after CsA treatment (Figure 7E).

We conclude that exposure to CsA and CsH leads to IFITM3 degradation only in cells expressing low/endogenous levels of this protein and only upon prolonged (>4 h) incubation, while ectopic overexpression or interferon stimulation render the protein resistant to degradation by these drugs. Regardless, CsA and CsH relocalize IFITM3 toward the Golgi and rescue virus–cell fusion in both low- and high-IFITM3-expressing cells, further supporting the IFITM3 relocalization-based mechanism of IFITM antagonism.

### 3.5. Rapamycin Induces IFITM Redistribution and Rescues Viral Fusion

Previous studies have reported the ability of rapamycin to interfere with IFITM3′s antiviral activity upon prolonged (>4 h) treatment of cells and concluded that this effect was due to the degradation of this protein via a lysosomal pathway [59,60]. The pretreatment of IFITM1- and IFITM3-expressing (or mock-transduced) A549 cells with rapamycin for 1.5 h enhanced IAVpp–cell fusion measured by the BlaM assay, albeit not as potently as with CsA (Appendix A). More importantly, rapamycin also redistributed IFITM3 toward the Golgi apparatus in A549.IFITM3 cells, although this distribution was not as pronounced as after CsA treatment (compare Appendix A and Figure 3, Figure 4 and Figure 5). In line with previous studies [59,60], Bafilomycin A1 (BafA1), a drug that prevents the acidification of endosomes and lysosomes [80], largely prevented IFITM3 degradation (Appendix A) upon rapamycin and also cyclosporine treatment in HeLa cells. Although the effect of cyclosporines and rapamycin on IFITM1 degradation was milder than on IFITM3, we still observed lower IFITM1 degradation in BafA1-treated cells (Appendix A). Treatment with NH_4_Cl (a lysosomotropic agent) also prevented IFITM degradation, further supporting the role of a lysosomal pathway in cyclosporine- and rapamycin-mediated IFITM degradation (Appendix A). Strikingly, MG-132 (a proteasome inhibitor) also inhibited IFITM1 and IFITM3 degradation (Appendix A), suggesting the involvement of a proteasomal degradation pathway in the regulation of IFITM expression levels.

Together, our data suggest that CsA, CsH, and rapamycin can quickly translocate IFITMs to the Golgi and thereby rescue viral fusion.

## 4. Discussion

Our results reveal a novel mechanism for the cyclosporine A and H antagonism of the antiviral activity of IFITM1 and IFITM3 through a redistribution of these proteins from the PM and endosomes, respectively, toward the Golgi apparatus. Similarly, rapamycin also induces a redistribution of IFITM3 toward the Golgi, albeit less potently than cyclosporines. The IFITMs’ relocation to the Golgi is relatively quick (under 1 h), leading to a cell-type-independent recovery of the VSV G and IAV HA pseudovirus fusion with IFITM-expressing cells. The lack of a significant effect of cyclosporine on virus fusion with IFITM-negative cells implies that fusion is enhanced through suppressing the IFITM’s antiviral activity and not through a non-specific fusion-enhancing effect of these drugs. Although prolonged incubation (>4 h) with cyclosporines does induce a degradation of IFITMs in cells expressing low/endogenous levels of these proteins, no notable reduction in the IFITM level is detected at the time of their relocalization toward the Golgi. Moreover, prolonged CsA/CsH treatment does not result in IFITM degradation in cells overexpressing these proteins or parental cells pretreated with interferon. It is also worth noting that blocking de novo protein synthesis with cycloheximide demonstrates that IFITM3 is not degraded within a few hours under basal conditions (Appendix A). Collectively, these findings strongly imply that the primary mechanism for rescuing the viral infection in IFITM-expressing cells pretreated with cyclosporines or rapalogs is the sequestration of these proteins in the Golgi area and not IFITM degradation, as proposed by previous studies [60,61]. 

In agreement with the observed IFITM relocalization to the perinuclear area under our experimental conditions, previous studies have reported an increased colocalization with endolysosomal markers after pretreatment with cyclosporine H or rapalog [60,61], but did not examine colocalization with the Golgi markers. We note, however, that the IFITM accumulation in the perinuclear space observed in these studies was not as pronounced as in our experiments. The differences in the IFITM distribution may be due to differences in the immunostaining and/or cell pretreatment protocols. 

How cyclosporines induce IFITM redistribution to the Golgi is not understood. Clearly, at least two effects are required for the observed phenotype—quick IFITM transport to and their entrapment within the Golgi. Without blocking IFITM’s exit from the Golgi, its colocalization with this organelle would not be so pronounced, as the proteins would continuously redistribute to the PM and endosomes. A recent work has identified a motif within the cytoplasmic intracellular loop (CIL) of IFITM3 that is involved in its export from the Golgi [81]. Mutations in this motif diminish IFITM3′s antiviral activity [82] and trap it in the Golgi, causing a “traffic jam” for other glycoproteins. The authors also show that overexpression of wild-type IFITM3 tends to impair its exit from the Golgi. It is thus likely that IFITM entrapment in the Golgi is largely a result of quick redistribution from the PM or endosomes, which greatly increases the IFITM concentration in the Golgi. 

What drives IFITM relocalization to the Golgi? The fact that CsH, which does not bind cyclophilins, is nearly as potent as CsA in driving the IFITM redistribution, rules out the involvement cyclophilins and calcineurin inhibition [72]. Many proteins undergo retrograde transport from early and late endosomes to the trans-Golgi. At least three pathways for this transport have been described: retromer, AP-1, and Rab9 (reviewed in [83,84,85]). It is possible that, in the presence of CsA, IFITM1 is internalized into early endosomes before being transported to the Golgi, consistent with its delayed accumulation in the Golgi compared to IFITM3 (Figure 5). All three above pathways are controlled by different proteins, but in many cases their functions and cargo overlap. Presently, it is unclear which pathway is responsible for IFITMs’ transport to the Golgi and whether these proteins accumulate in cis- or trans-Golgi. Further experiments involving biochemical assays and/or super-resolution microscopy are needed to identify the IFITM-enriched compartment(s) in cells treated with cyclosporines. 

Recent studies revealed that, besides their antiviral activity, IFITMs can negatively impact key cellular functions. IFITM overexpression has been implicated in cancer [86,87], the disruption of trophoblast fusion required for placenta formation [20], and the inhibition of endosomal transport/fusion, including the back-fusion of intraluminal vesicles with a limiting membrane of multivesicular bodies [88]. Additionally, the endogenous expression of IFITMs in stem cells diminishes the efficiency of lentivirus-based gene therapy [60,61]. Pretreatment with cyclosporines allows for a quick and nontoxic removal of IFITMs from their respective localization sites, thereby rendering the cells much more susceptible to viral transduction. It is therefore important to understand processes that regulate IFITM transport and degradation in cells to be able to lessen their adverse effects, potentiate their antiviral activity, and increase the spectrum of restricted viruses through controlling their sub-cellular localization and expression levels. Regardless of the exact mechanism of IFITM retention, our results demonstrate the importance of IFITM trafficking through cellular compartments for their antiviral activity and implicate yet unknown host factors in regulating the transport of these restriction factors. Our future studies will aim to identify host factors/processes responsible for the effects of cyclosporines on IFITMs’ subcellular localization. 

## Figures and Tables

**Figure 1 biomolecules-13-00937-f001:**
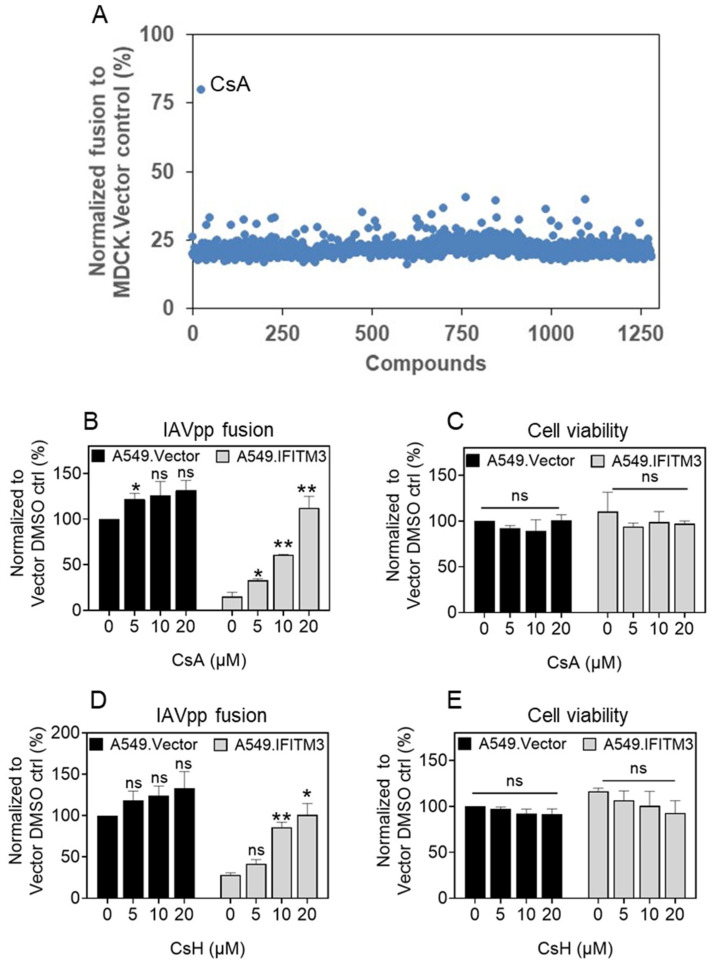
Cyclosporines A and H inhibit IFITM3 antiviral activity. (**A**) Scatter plot of screening results of 1280 compounds LOPAC library at 20 µM final concentration. The screening was performed using BlaM assay for IAVpp fusion in MDCK.IFITM3 cells seeded in 384-well plates. The results were normalized to IAVpp fusion in MDCK.Vector cells used as a control for efficient fusion of IAVpp (see also Appendix A). Dose–response on IAVpp fusion and cell viability for CsA (**B**,**C**) and CsH (**D**,**E**) in A549.Vector and A549.IFITM3 cells. Target cells were treated with cyclosporines or DMSO for 1.5 h at 37 °C prior to IAVpp binding. Data are means and S.D. of two independent combined experiments, each performed in triplicate. Statistical analysis was caried out using Student’s *t*-test. * *p* < 0.05; ** *p* < 0.01; ns, not significant.

**Figure 3 biomolecules-13-00937-f003:**
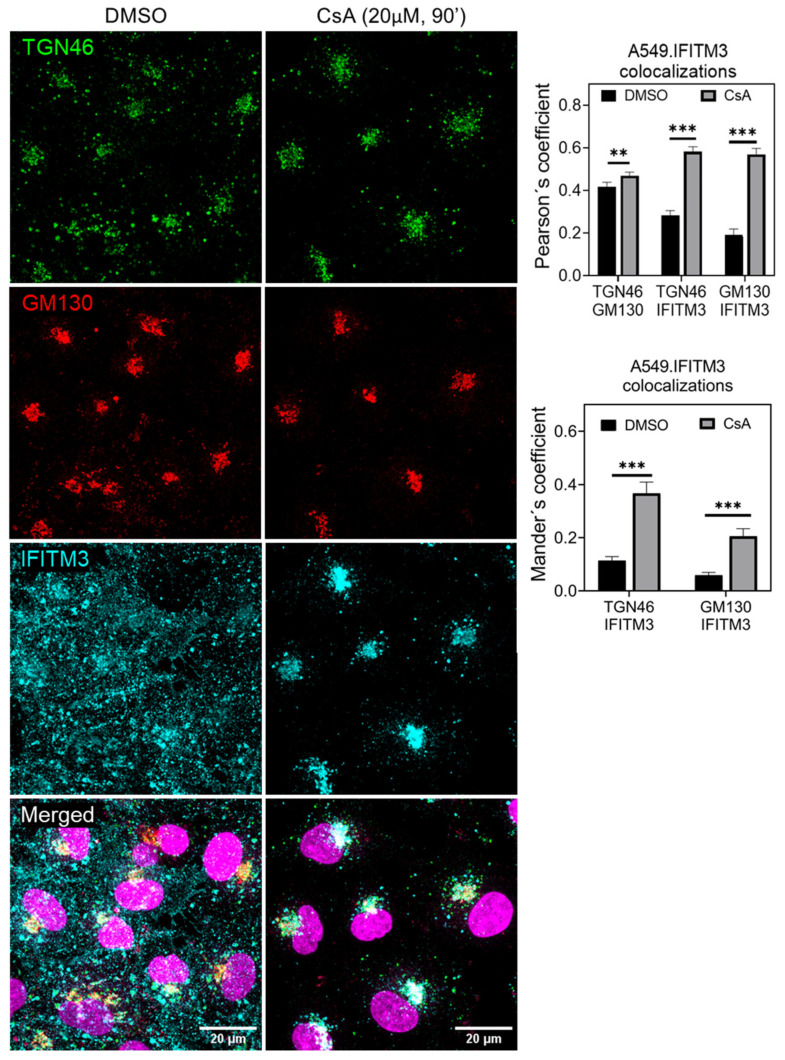
Cyclosporine A relocates IFITM3 toward the Golgi apparatus. A549-IFITM3 cells were incubated with DMSO or CsA (20 µM) for 1.5 h, fixed, permeabilized, and stained for respective proteins. For each sample, 5 image fields were acquired. The JaCoP ImageJ plugin was used to calculate IFITM3, TGN46, and GM130 colocalization using both Pearson’s and Mander’s coefficient. Statistical analysis was performed using Student’s *t*-test. **, *p* < 0.01; *** *p* < 0.001. See also Appendix A.

**Figure 4 biomolecules-13-00937-f004:**
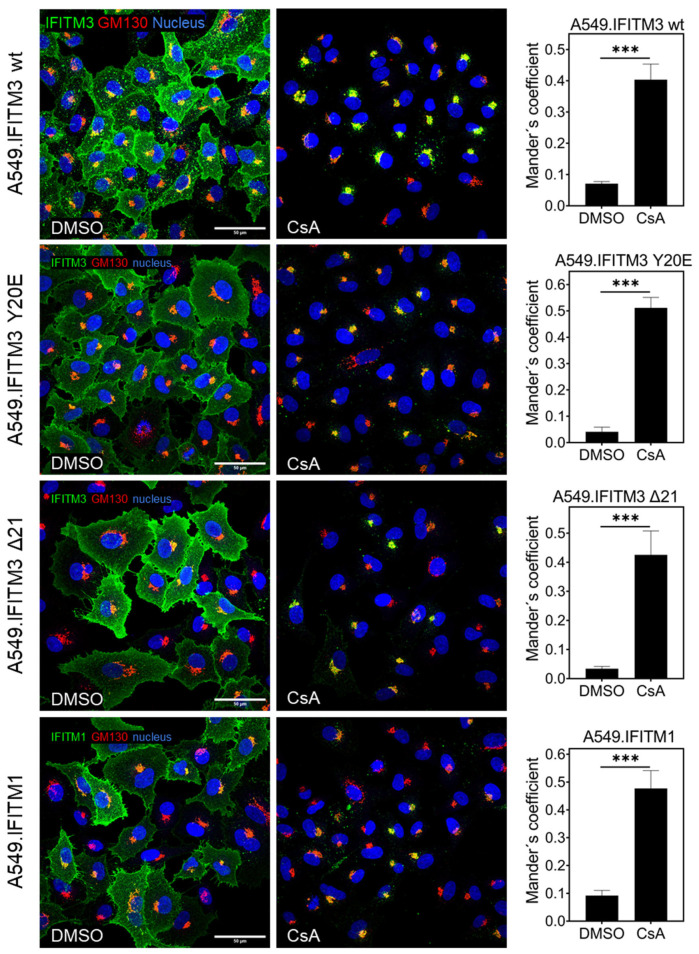
Cyclosporine A induces IFITM3 relocalization irrespective of the N-terminal endosome-sorting signal. A549 cells transduced with wild-type IFITM3, IFITM3 Y20E, IFITM3 NΔ21, or IFITM1 were incubated with CsA (20 µM) for 1.5 h, fixed, and stained for respective protein together with Golgi marker GM130. For each sample, 3 image fields were acquired. The JaCoP ImageJ plugin was used to measure respective IFITM and GM130 colocalization. Since expression of IFITM1 and IFITM3 mutants was not homogenous across the acquired images, we decided to compare IFITM and GM130 colocalization using Mander’s correlation (% of IFITM signal overlapping with GM130 signal) instead of Pearson’s correlation (relationship between signals) for this and other images. Statistical analysis was performed using Student’s *t*-test. *** *p* < 0.001.

**Figure 5 biomolecules-13-00937-f005:**
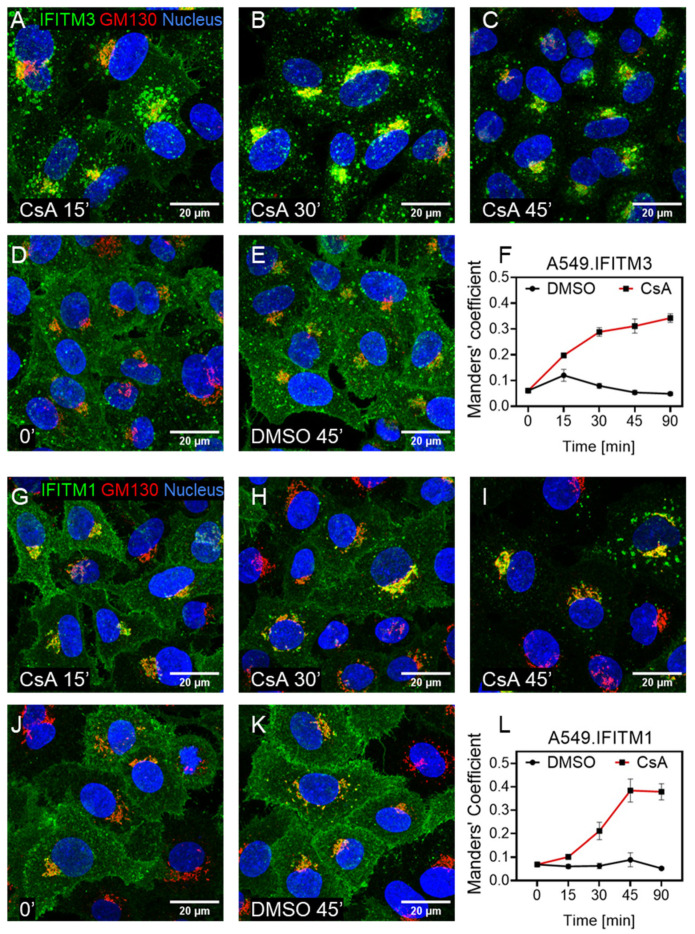
Cyclosporine-A-mediated IFITM3 relocalization toward the Golgi is completed within 45 min. A549 cells transduced with either IFITM3 or IFITM1 were incubated with CsA (20 µM) for indicated time, fixed, and stained for respective protein together with Golgi marker GM130. (**A**–**E**) Images of IFITM3 and Golgi signals after the indicated time of pretreatment with CsA or DMSO (control). (**F**) Analysis of colocalization of IFITM3 and Golgi markers from panels A–E. (**G**–**K**) Images of IFITM1 and Golgi signals after the indicated time of pretreatment with CsA or DMSO (control). (**L**) Analysis of colocalization of IFITM3 and Golgi markers from panels G–K. For each sample, 5 image fields were acquired. The JaCoP ImageJ plugin was used to measure respective IFITM and GM130 colocalization.

**Figure 6 biomolecules-13-00937-f006:**
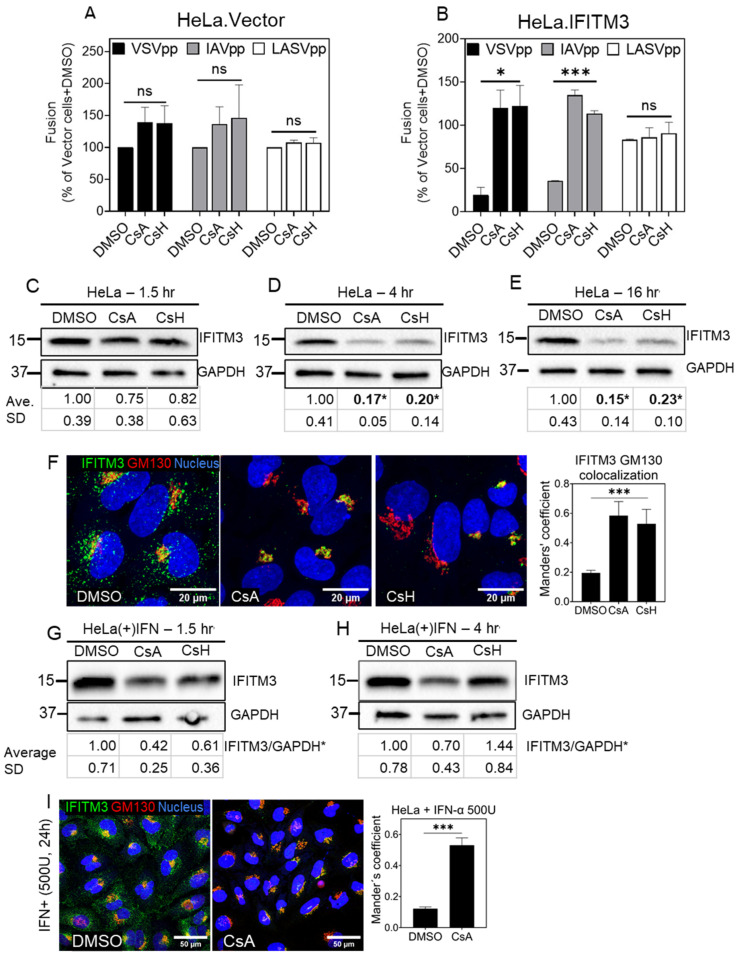
Cyclosporine A induces IFITM3 degradation in HeLa cells. HeLa.Vector (**A**) and HeLa.IFITM3 (**B**) cells were incubated with DMSO or 20 µM CsA, or CsH for 1.5 h before adding BlaM-Vpr containing pseudoparticles for virus–cell fusion measurements. Data are means and S.D. of two independent combined experiments, each performed in triplicate. Statistical significance was assessed using one-way ANOVA. For Western blot analysis, HeLa cells were incubated with DMSO or 20 µM CsA or CsH for 1.5 h (**C**), 4 h (**D**), or 16 h (**E**), and total cell lysate was analyzed. For immunostaining analysis, HeLa cells were incubated with DMSO or 20 µM CsA or CsH for 1.5 h, fixed, permeabilized, and stained for IFITM3 (**F**). For each sample, 3 image fields were acquired. The JaCoP ImageJ plugin was used to measure respective IFITM and GM130 colocalization. HeLa cells were treated with 500 units of interferon-alpha for 24 h followed by DMSO, CsA, or CsH treatment for 1.5 h (**G**) or 4 h (**H**) and whole-cell lysate was analyzed by Western blotting. Densitometry analysis was performed using ImageJ for three independent experiments. Statistical significance was determined using Student’s *t*-test. ***** *p* < 0.05. For IFN-treated HeLa cells, only duplicates were used for this analysis. (**I**) HeLa cells were treated as stated above, fixed, and stained for IFITM3 and GM130. For each sample, 3 image fields were acquired. The JaCoP ImageJ plugin was used to measure respective IFITM and GM130 colocalization. Statistical analysis was performed using Student’s *t*-test. * *p* < 0.05; *** *p* < 0.001; ns, not significant. See also Appendix A.

**Figure 7 biomolecules-13-00937-f007:**
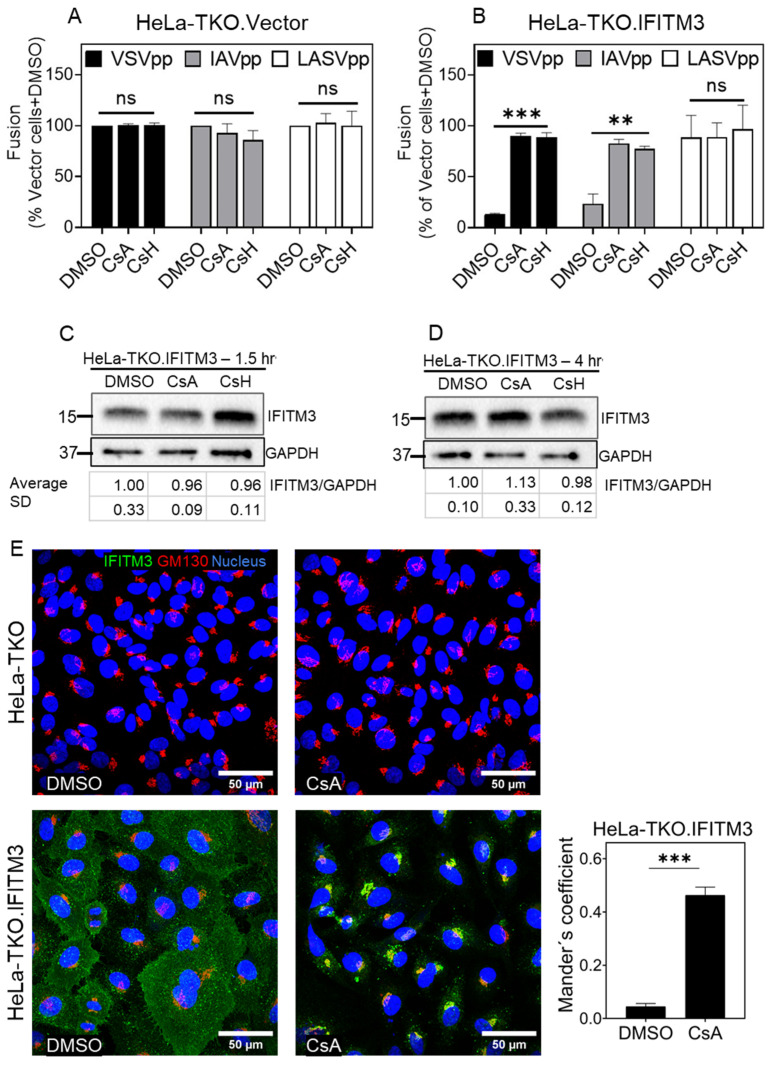
Cyclosporine-A-driven rescue of viral fusion is through IFITM3 antagonism. HeLa-TKO.Vector (**A**) and HeLa-TKO.IFITM3 (**B**) cells were incubated with with DMSO or 20 µM CsA or CsH for 1.5 h before adding BlaM-Vpr containing pseudoparticles for virus–cell fusion measurements. Data are means and S.D. of two independent combined experiments, each performed in triplicate. Statistical significance was assessed using one-way ANOVA. For Western blots, HeLa-TKO.IFITM3 was incubated with DMSO or 20 µM CsA or CsH for 1.5 h (**C**) or 4 h (**D**), and total cell lysate was analyzed. The mean-normalized band intensities and S.D. from three independent experiments are shown under the gels. (**E**) HeLa-TKO and HeLa-TKO.IFITM3 cells were treated as indicated previously, fixed, and stained for IFITM3 and GM130. For each sample, 3 image fields were acquired. The JaCoP ImageJ plugin was used to measure respective IFITM and GM130 colocalization. Statistical analysis was performed using Student’s *t*-test. ** *p* < 0.01; *** *p* < 0.001; ns, not significant.

## Data Availability

All pertinent data are included in this paper.

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
