# Peer review of "Cyclosporines Antagonize the Antiviral Activity of IFITMProteins by Redistributing Them toward the Golgi Apparatus"

_biomolecules, 2023, doi:10.3390/biom13060937_

Round 1
Reviewer 1 Report
In this manuscript, Prikryl and collaborators have investigated the mechanisms by which cyclosporines A and H antagonize the antiviral activity of IFITM proteins (i.e., IFITM3 and IFITM1) on virus fusion of different viruses, such as VSV and IAV, using pseudotyped HIV-1 particles and a previously described BlaM assay. While it has been previously reported that cyclosporines antagonized the antiviral activity of IFITM proteins through a mechanism related to their degradation, the authors report here, mainly using A549 cells ectopically stably expressing IFITM proteins, but also HeLa cells constitutively expressing low level of those proteins, interesting and very convincing immunofluorescence data showing that cyclosporines can also provoke a fast and massive relocalization (in less than 1.5 h) of IFITM1 and 3 from the plasma membrane and endosomes, respectively, toward the Golgi apparatus. Similarly, rapamycin was also able to induce the relocalization of IFITM proteins toward the Golgi compartment. In contrast, they were able to detect cyclosporine-induced degradation of IFITM1 and 3 only in cells expressing low levels of endogenous proteins such HeLa cells after 4 h of cyclosporine treatment. By contrast, there was no apparent degradation detected in HeLa cells previously stimulated by IFN treatment for induction of IFITM protein expression.
Even if all the experiments were performed in cell-lines, and should be further characterize in primary cells, the interesting data reported throughout this manuscript strongly support that cyclosporines can also induce relocalization of IFITM proteins from their sites of action and thus inhibit their antiviral activity. While the two mechanisms of cyclosporines (degradation versus intracellular redistribution) on IFITM proteins are not mutually exclusive, the authors could even speculate in the Discussion Section about the possibility of a two-step mechanism in which cyclosporines first provoke the redistribution of IFITM proteins to the Golgi apparatus where they will be then degraded in a proteasomal-dependent manner, since they show on Fig. S5 that the MG132 proteasome inhibitors was able to revert the degradation of IFITM proteins induced by cyclosporines or rapamycine.
Just a few minor points can be mentioned:
1) Figure 2: There is maybe a mistake in the statistical analysis reported on panel A regarding the effect of cyclosporines on fusion of virus particles pseudotyped with the Lassa virus envelope?
2) Figure 3. Because there is a significant increase of the co-localization coefficient regarding TGN46 and GM130 staining (on upper right part of the figure), I am wondering whether cyclosporine A treatment could modify the distribution of Golgi markers, and thus disturb the organization and function of the Golgi apparatus?
3) Figure 6. It is not clear whether wild-type HeLa cells, not overexpressing IFITM3, were used in panels C-F of the figure? In addition, the authors could show both in Western blot and immunofluorescence analyses how IFN treatment induces IFTM protein expression (panels G-I), and controls on non-treated cells could be shown.
Author Response
Reviewer 1:
In this manuscript, Prikryl and collaborators have investigated the mechanisms by which cyclosporines A and H antagonize the antiviral activity of IFITM proteins (i.e., IFITM3 and IFITM1) on virus fusion of different viruses, such as VSV and IAV, using pseudotyped HIV-1 particles and a previously described BlaM assay. While it has been previously reported that cyclosporines antagonized the antiviral activity of IFITM proteins through a mechanism related to their degradation, the authors report here, mainly using A549 cells ectopically stably expressing IFITM proteins, but also HeLa cells constitutively expressing low level of those proteins, interesting and very convincing immunofluorescence data showing that cyclosporines can also provoke a fast and massive relocalization (in less than 1.5 h) of IFITM1 and 3 from the plasma membrane and endosomes, respectively, toward the Golgi apparatus. Similarly, rapamycin was also able to induce the relocalization of IFITM proteins toward the Golgi compartment. In contrast, they were able to detect cyclosporine-induced degradation of IFITM1 and 3 only in cells expressing low levels of endogenous proteins such HeLa cells after 4 h of cyclosporine treatment. By contrast, there was no apparent degradation detected in HeLa cells previously stimulated by IFN treatment for induction of IFITM protein expression.
Even if all the experiments were performed in cell-lines, and should be further characterize in primary cells, the interesting data reported throughout this manuscript strongly support that cyclosporines can also induce relocalization of IFITM proteins from their sites of action and thus inhibit their antiviral activity. While the two mechanisms of cyclosporines (degradation versus intracellular redistribution) on IFITM proteins are not mutually exclusive, the authors could even speculate in the Discussion Section about the possibility of a two-step mechanism in which cyclosporines first provoke the redistribution of IFITM proteins to the Golgi apparatus where they will be then degraded in a proteasomal-dependent manner, since they show on Fig. S5 that the MG132 proteasome inhibitors was able to revert the degradation of IFITM proteins induced by cyclosporines or rapamycine.
Response: We thank the reviewer for the positive comments and agree that verifying the CsA effect in primary cells should be the future direction.
Just a few minor points can be mentioned:
1) Figure 2: There is maybe a mistake in the statistical analysis reported on panel A regarding the effect of cyclosporines on fusion of virus particles pseudotyped with the Lassa virus envelope?
Response: We do observe some enhancement of LASVpp fusion and a modest (but not significant) enhancement of IAVpp fusion with CsA-treated A549 cells. This is likely due to non-IFITM-related effects of CsA on cells.
2) Figure 3. Because there is a significant increase of the co-localization coefficient regarding TGN46 and GM130 staining (on upper right part of the figure), I am wondering whether cyclosporine A treatment could modify the distribution of Golgi markers, and thus disturb the organization and function of the Golgi apparatus?
Response: We thank the reviewer for this comment. We do not observe obvious changes in the shape/size distributions of the Golgi markers used. However, we cannot rule out subtle changes in the Golgi apparatus or other cellular structures under these conditions. Since assessing subtle changes in the Golgi structure would require much finer tools and major time investment and because the focus of this work is on the IFITM redistribution, we would prefer to address a possible CsA effect on the Golgi in the future.
3) Figure 6. It is not clear whether wild-type HeLa cells, not overexpressing IFITM3, were used in panels C-F of the figure? In addition, the authors could show both in Western blot and immunofluorescence analyses how IFN treatment induces IFTM protein expression (panels G-I), and controls on non-treated cells could be shown.
Response: We used parental HeLa cells in these experiments, as was indicated in both figures and the manuscript. The results of independent experiments with IFN-untreated HeLa cells were shown in panels C-F. We added new Supplemental Figure S4 to better illustrate the effect of IFN treatment on IFITM3 expression in HeLa cells and the effect of CsA on subcellular distribution of IFITM3.
Reviewer 2 Report
In this report Prikryl and colleagues describe experiments that first searched in a large library of compounds for inhibitors of interferon induced transmembrane proteins inhibition of fusion, concentrating on IFITM3. The authors identified Cyclosporin A (and then its analog Cys H) as mediating significant inhibition of effect of IFTIM3 and provide evidence that Cys A promotes translocation of IFTTM3 from the plasma membrane to the Golgi membranes, thus moving it away from the site of viral-cell fusion with some viruses, particularly Influenza A. Evidence is also provided that the effect is not due to degradation of Cys A, at least under conditions/cells where it is expressed robustly. The experiments indicating that the IFITM3 inhibition by rapamycin is mediated by a similar mechanism.
Overall, this is a very careful and well-documented study. My only suggestions are that some of the experimental figures supplied as supplemental (for example the Ampho B experiments in Fig S2) should be included in the manuscript rather than as supplemental material. If the manuscript is then considered too long, the HeLa experiments (final section) could be mentioned. It is not clear why there is degradation under conditions of low IFTIM3 expression whereas there is no degradation with high expression.
Author Response
Reviewer 2:
In this report Prikryl and colleagues describe experiments that first searched in a large library of compounds for inhibitors of interferon induced transmembrane proteins inhibition of fusion, concentrating on IFITM3. The authors identified Cyclosporin A (and then its analog Cys H) as mediating significant inhibition of effect of IFTIM3 and provide evidence that Cys A promotes translocation of IFTTM3 from the plasma membrane to the Golgi membranes, thus moving it away from the site of viral-cell fusion with some viruses, particularly Influenza A. Evidence is also provided that the effect is not due to degradation of Cys A, at least under conditions/cells where it is expressed robustly. The experiments indicating that the IFITM3 inhibition by rapamycin is mediated by a similar mechanism.
Overall, this is a very careful and well-documented study. My only suggestions are that some of the experimental figures supplied as supplemental (for example the Ampho B experiments in Fig S2) should be included in the manuscript rather than as supplemental material. If the manuscript is then considered too long, the HeLa experiments (final section) could be mentioned. It is not clear why there is degradation under conditions of low IFTIM3 expression whereas there is no degradation with high expression.
Response: We thank the reviewer and agree that the results could be split differently between the main and supplemental figures but would prefer to keep the current figure arrangement, simply because the Ampho B effect is not the focus of this study.